# The Inc FII Plasmid and its Contribution in the Transmission of *bla_NDM-1_* and *bla_KPC-2_* in *Klebsiella pneumoniae* in Egypt

**DOI:** 10.3390/antibiotics8040266

**Published:** 2019-12-13

**Authors:** Eman Ramadan Mohamed, Mamdouh Yones Ali, Nancy G F M Waly, Hamada Mohamed Halby, Rehab Mahmoud Abd El-Baky

**Affiliations:** 1Department of Microbiology and Immunology, Faculty of Pharmacy, Al-Azhar University, Assuit 11651, Egypt; eman-ramadan86@aun.edu.eg (E.R.M.); mamdouhyones75@yahoo.com (M.Y.A.); hamadahalby@ymail.com (H.M.H.); 2Department of Microbiology and Immunology, Faculty of Pharmacy, Minia University, Minia 61519, Egypt; nancy.gamil1@mu.edu.eg; 3Department of Microbiology and Immunology, Faculty of Pharmacy, Deraya University, Minia 61519, Egypt

**Keywords:** carbapenemase-producing *Klebsiella pneumoniae*, *bla_KPC-2_*, *bla_NDM-1_*, plasmid replicon type hvCPKP, PBRT, MβL, mCIM, eCIM

## Abstract

The emergence of *bla_KPC-2_* and *bla_NDM-1_* producing *Klebsiella pneumoniae* represents a great problem in many Egyptian hospitals. One hundred and twenty-six *K*. *pneumoniae* isolates from patients admitted to Assiut University Hospital were identified by an API20E kit. Carbapenemase-producing *K*. *pneumoniae* (CPKP) was detected by the modified carbapenem inactivation method (mCIM), the EDTA-modified carbapenem inactivation method (eCIM), and an E-test. Based on the polymerase chain reaction, all isolates were negative for *bla-_VIM-1_* and *bla-_IMP-1_*, fifteen of these isolates were positive for both *bla_KPC-2_* and *bla_NDM-1_*, two isolates were positive for *bla_KPC-2_* only, and twenty-eight isolates were positive for *bla-_NDM-1_* only. Although one isolate was positive for the string test, all CPKP isolates were negative for capsular genes. Only 71.1% of CPKP transferred their plasmids to their corresponding transconjugants (*E. coli* J53). The resistance patterns of the clinical isolates and their transconjugates were similar, except for 12 isolates, which showed differences with their transconjugates in the resistance profile of four antibiotics. Molecular typing of the plasmids based on replicon typing showed that Inc FIIK and FII plasmids predominated in isolates and their transconjugants carrying *bla_KPC-2_* and/or *bla_NDM-1_.* Conjugative Inc FII plasmids play an important role in the spread of CPKP, and their recognition is essential to limit their spread.

## 1. Introduction

*K. pneumonia* is a common bacterial pathogen that can cause several life-threatening infections, such as blood stream infections, pneumonia, as well as urinary tract, post-surgical, and intensive care-related infections [1,2]. Such infections result in significant morbidity and mortality. *K. pneumonia* also significantly increases medical hospital costs [3]. It can spread well and survive in the hospital environment and frequently causes nosocomial infections and outbreaks [4]. *K. pneumonia*’s major route of infection is aspiration of the oropharyngeal secretions of the patients colonized by it. The gastrointestinal tract of patients may also play a secondary role. Further, contaminated medical equipment and the hands of health care workers may contribute to sustaining colonization and creating an increased risk of infection [5]. Additionally, antibiotic therapy can suppress normal bacterial flora and lead to an over-growth of multidrug resistant *K. pneumoniae* [6]. 

The increasing incidence of carbapenemase-producing *K. pneumoniae* (CPKP) has become a great challenge for infection control in human health and has resulted in treatment failure [7]. *K. pneumoniae* was found to harbor plasmid-mediated carbapenemase of different classes, such as Class A (e.g., KPC), Class B as Verona integron-encoded MBL (VIM), imipenemase (IMP) and New Delhi-metallo beta-lactamase (NDM) or Class D (e.g., oxacillinases (OXA) enzymes [8,9]. Carbapenem resistant *K. pneumoniae* was reported and isolated from many Egyptian hospitalized patients [10,11,12]. Furthermore, the co-existence of carbapenems and other drug-resistance determinants, such as aminoglycoside and quinolone resistance genes in *K. pneumoniae*, was found to be a great challenge in treating infections caused by this pathogen [8].

Large conjugative plasmid-carrying genes that encode resistance were found to have a great role in the spread and acquisition of resistance among *K. pneumoniae strains* [13,14]. The molecular typing of resistance plasmids based on replicon typing is currently applied to study the acquisition and spread of antimicrobial resistance in *K. pneumoniae* [15]. Carbapenemase-producing *K. pneumoniae* (KPC) and New Delhi-metallo beta-lactamase (NDM) are the most noteworthy carbapenemase-encoding genes that are found to enable the spread of these resistance genes in transferable plasmids. Thus, tracing the source of endemic plasmids that could be responsible for the spread of a resistance trait among strains is the backbone for the development of efficient treatment policies, as well as applicable infection control measures [16,17,18,19]. Treating infections with antibiotics that are inactive against the causative agent leads to increasing the selective pressure for nosocomial organisms to express antibiotic resistance and dissemination and the persistence of an endemic plasmid among strains [20]. This means that the withdrawal of ineffective drug, earlier correct drug selection, or using effective combinations with reliable spectra of activity and effective treatments against these pathogens may stop selection pressure and interfere with the transmission of the plasmid [21,22]. Although many studies have recently reported the emergence of *K. pneumoniae* harboring NDM or KPC in Egypt, limited studies are available on plasmids responsible for transferring resistance determinants in multidrug resistant *K. pneumoniae* in this country. As a result, our aim was to study and identify the plasmid type responsible for the transmission of *blaNDM-1* and *blaKPC-2* in *K. pneumoniae* in Egypt in order to trace the transfer of this plasmid among *K. pneumoniae* strains [11].

## 2. Results

### 2.1. Cabapenemase Production by K. pneumoniae 

One hundred and twenty-six *K. pneumoniae* strains isolated from different departments at Assiut University Hospital (April 2018 to May 2019) were tested for their production of carbapenemase enzymes using mCIM. Carbapenemase activity was found in 77/126 (61.1%) of the *K. pneumoniae* isolates. *K. pneumoniae* isolates positive for mCIM were tested for Metallo-β-lactamase production using the eCIM method. Out of 77 carbapenemase-producing *K. pneumoniae*, 51 isolates (66.2%) were positive for eCIM. In addition, it was observed that 61/77 of carbapenemase producing *K. pneumoniae* were imipenem resistant, with MIC of ≥4 μg/mL.

### 2.2. Hypermucoviscocity Phenotyping

Only one isolate (Kp27) showed the hypermucoviscocity phenotype (Positive string test).

### 2.3. Antimicrobial Resistance Pattern of Carbapenemase Producing K. pneumoniae (CPKP)

The antibacterial susceptibility profile was tested for all 77 CPKP isolates. It was found that Carbapenemase producing *K. pneumoniae* were completely resistant to Aztreonam, Amoxicillin, Amox-Clavulanic acid, Piperacillin, Ceftriaxone, Ceftazidime, Cefoperazone, Cefpodoxime, and Meropenem but showed high resistance to the other tested antibiotics (Figure 1).

### 2.4. PCR for Carbapenemase Genes and KI and K2 Capsular Genes 

All *K. pneumoniae* isolates were tested for carbapenemase genes and K1 and K2 capsular genes. It was found that all non CPKP were negative for these genes. On the other hand, Carbapenemase-encoding genes were founded in 45/77 (58.44%) of carbapenemase-producing *K. pneumoniae* isolates. *Bla_NDM-1_* was present in 43/77 (55.84%) isolates and *bla_KPC-2_* in 17/77 (22.07%) isolates. 

All CPKP isolates were negative for *bla_VIM-1_*, *bla_IMP-1_*, *wzy_K1*, and *wzy_K2* capsular genes (Figure 2). Among the isolates tested positive for carbapenemase genes, fifteen (15/45, 33.3%) were positive for both *bla_NDM-1_* and *bla_KPC-2_.* Two isolates (2/45, 4.4%) were positive for *bla_KPC2_* only and twenty-eight isolates (28/45, 62.2%) for *bla_NDM-1_* only. 

### 2.5. Comparison of Phenotypic and Genotypic Methods for Detection of CPKP

Isolates were divided into four groups according to their positivity for the tested genes (Table 1). The isolates negative for all genes (Group 1) were negative for mCIM. Out of the 28 isolates (Group 2) that tested positive for *bla_NDM-1_* and mCIM, nine isolates were metallo-β-lactamases (MBL) producers (positive for eCIM), while the two isolates (KP8, KP17) positive for *blaKPC-2* (Group 3) were positive for the mCIM and eCIM test (Table 1). In addition, the five isolates of Group 4, which were positive for *bla_NDM1_* and *bla_KPC2_* and mCIM, were positive for eCIM and were considered to be MBL producers.

### 2.6. Plasmid Transfers

Conjugation experiments were carried out for all *bla_KPC-2_* and/or *bla_NDM-1_* carrying *K. pneumoniae* isolates (n = 45). Only 32 of these were found to have the ability to transfer their plasmids by conjugation to their corresponding transconjugants (*E. coli J53*). These transconjugants were tested for *bla_KPC-2_* and/or *bla_NDM-1_* genes by the PCR method. All thirty-two transconjugants were found to harbor *bla_KPC-2_* and/or *bla_NDM-1_* genes as their parent cells.

All transconjugants were observed to exhibit Multi-drug resistance (MDR) phenotypes like those of the donor *K. pneumoniae* isolates, except 12 of the transconjugants were found to have different antibiotic resistance profiles. The difference in the resistance profiles among these isolates was observed with the following antibiotics: Tetracycline, Gentamicin, Sulfamethoxazole/trimethoprim, and Levofloxacin. The resistance profiles of the 12 tested isolates and transconjugates against these four antibiotics are listed in Table 2. 

### 2.7. PCR-Based Replicon Typing of bla_KPC2_ and/or bla_NDM1_ Encoding K. pneumoniae

PCR-based replicon typing (PBRT) showed typing for only 29/32 of the self-transmissible bla_KPC-2_ and/or bla_NDM-1_ encoding *K. pneumoniae* isolates. The replicon types identified were FIIK, FII, FIB, FIC, L, and Inc M. It was found that FIIK and FII were highly distributed among *bla_KPC-2_* and/or *bla_NDM-1_* encoding *K. pneumoniae* isolates, followed by the FIB, FIC, Inc L, and Inc M groups. Furthermore, PBRT failed to detect the replicon types of the plasmids of the remaining three isolates (Kp5, Kp6, Kp22) (Table 3).

### 2.8. Transfer of Plasmids with a Specific Replicon Type to the Transconjugants

PCR-based replicon typing of the transconjugants (*E. coli J53*) carrying *bla_KPC-2_* and/or *bla_NDM-1_* encoding genes (n = 32) revealed that nine (28.1%) of the strains were positive for Inc FIIK alone, while three (9.3%) of the transconjugants were positive for all FII, FIIK, and FIB incompatibility types. Two (6.25%) of the strains carried plasmids belonging to the Inc FII group only, whereas Inc FIB was found in one (3.1%) of them. On the other hand, PBRT confirmed the transfer of Inc FII and FIIK plasmids to two transconjugants and the transfer of Inc FIIK and FIB groups to other two transconjugants. The remaining strains (*n* = 13) could not be typed by the PBRT method (Table 3). Also, it was observed that Inc L, M, and FIC plasmids did not show the ability to transfer to the recipient cells.

## 3. Discussion

Carbapenemase-producing *Klebsiella pneumoniae* (CPKP) is one of the most widespread pathogens among hospital-acquired multi-drug resistant pathogen associated infections [7]. Although many reports exhibited the emergence of carbapenem resistant *K. pneumoniae* in Egypt [12,23] in the last decades, the data on plasmid epidemiology and which plasmid types are mainly associated with the spread of *bla_KPC2_* and/or *bla_NDM1_* plasmids in Egypt are rare. Thus, in this study, we attempted to determine the plasmids responsible for transferring the determinants of carbapenem-resistance among *K. pneumoniae* isolates in this country.

In the present study, carbapenemase phenotypic activity was detected in 77/126 (61.1%) of the *K. pneumoniae* isolates by an mCIM test. Although the rate we observed was lower than that in a study in the USA [24], it was higher than the previous reports in Egypt. The prevalence of CPKP was found to be 33.3% among *K. pneumoniae* isolates in a study performed by Moemen and Masallat [23]; 44.3% at the Suez Canal University Hospitals in a study performed by El-Sweify et al. [10]; and 13.9% in the Egyptian National Cancer Institute in a study performed by Ashour and El-Sharif [25]. Increasing rates of CPKP isolates may be due to the misuse of carbapenems, as carbapenems are considered the best choice for the treatment of serious infections caused by multi-drug resistant pathogens in our hospitals, where there is presently no implementation of an antimicrobial stewardship program [26].

Our results show that the mCIM negative isolates were also negative for both *bla_KPC2_* and/or *bla_NDM1_*, and there was compatibility between the results of the mCIM and PCR results among carbapenemase producing isolates. The sensitivity of this test was 100% compared to PCR, the gold standard test, suggesting that this test can be used as a primary test for the detection of CPKP. On the contrary, the eCIM test did not identify isolates that were positive for both *bla_KPC2_* and *bla_NDM1_* in accordance with the CLSI [27] report.

The capsular serotypes K1 and K2 of *K. pneumoniae* are the predominant virulent strains that contribute to the high mortality rate associated with *K. pneumoniae* infections [28]. Although there was only one isolate (Kp27) positive for the string test, all CPKP isolates were negative for the K1 and K2 capsular gene. This may be due to the fact that antibiotic resistance is not dependent from the hypervirulence of KP [29], and isolates were recovered from hospital-acquired infections, not from community-acquired infections [30,31]. In addition, the incidence of *bla_NDM1_* producers among the CPKP in our hospital (55.84%) was much higher than that previously reported by Yan et al. [8]. *Bla_NDM1_* was first detected in Egypt from one *K. pneumoniae* isolate in 2013 by Abdelaziz et al. [32] and then from two *Pseudomonas aeruginosa* isolates [33], as well as in *Acinetobacter baumannii* isolates [34], which indicates that this gene may spread among *Enterobacteriaceae* by transferable plasmids [35,36].

In Egypt, the plasmid types that are mainly associated with the spread of *bla_KPC2_* and/or *bla_NDM1_* have not been identified, except for the eight carbapenem-resistant NDM1-producing *K. pneumoniae* isolates carried by non-transferrable plasmids (either IncR or untypeable), which were reported by Gamal et al. [11].

Our study showed that most *bla_KPC2_* and/or *bla_NDM1_* plasmids were successfully transferred by conjugation and that *bla_KPC2_* and/or *bla_NDM1_* were commonly carried on the Inc FII plasmid. Similar results were reported by [37], who reported that *bla*_KPC-2_ was located on Inc FII plasmids, but in another study performed by Jin et al. [38], *bla_KPC2_* was found to be located on non-self-transmissible plasmids or on the chromosome.

Our results showed the predominance of both Inc FIIK and FII plasmids among the isolates and their transconjugants carrying *bla_KPC2_* and/or *bla_NDM1_*, suggesting that these types of plasmids mediate horizontal transmission and contribute to the dissemination of *bla_KPC2_* and/or *bla_NDM1_* in the environment of the Egyptian hospitals, which represents a great challenge and an important factor in the dissemination of resistance and treatment failure in cases of severe infections. Agyekum et al. [39] reported that FIIK is the most common plasmid replicon found in *K. pneumoniae*. Another study done by Al-Marzooq et al. [40] reported that the most common plasmid replicons were IncR and IncL/M. Moreover, a high incidence of conjugative FIIK (69%) and L/M plasmids (66%) found in CPKP isolates were reported in Saudi Arabia [41].

Based on the previous findings, the Inc F plasmid is the most frequently described plasmid type to be associated primarily with resistance genes in humans [42,43]. 

In our study, Inc L, M, and FIC plasmids were not detected in the recipient cells, indicating that these types of plasmid were non-conjugative, and their role in the spread of carbapenem resistance among CPKP isolates is not clear; it is possible that these plasmids have no role in the transfer of resistance in our area. The prevalence of certain plasmid types is different even among samples collected from different sources within the same city and between different countries [42]. 

The antibiotic sensitivity tests of transconjugants showed that all transconjugants were resistant to Levofloxacin-like donor cells, except for KP19 and KP25, which can be explained by the co-existence and the transfer of quinolone resistant genes with carbapenem resistant determinants in the same strain [8].

Based on the previous results, diverse clones of multidrug-resistant (MDR) *K. pneumoniae* can spread between patients and the environment in the same hospital, especially with the increase of the number of patients and the inappropriate application of infection control guidelines among hospitals in Egypt. Therefore, the horizontal transmission of carbapenem-resistant plasmids among admitted patients may encourage the dissemination of carbapenem resistance to other species of *Enterobacteriaceae*, thereby leading to the maintenance of carbapenem-resistant clones in patients and/or the environment [7,40]. As a result, it is necessary to limit the spread of CPKP carrying highly conjugative Inc F plasmid types within healthcare settings and to implement effective infection control measures.

To the best of our knowledge, this is the first report on the identification of the conjugative Inc F plasmid that circulates and spreads in NDM-1 and KPC-2-producing CPKP isolates from a single hospital in Egypt.

## 4. Methods

### 4.1. Klebsiella pneumoniae Isolation and Identification

One hundred and twenty-six *Klebsiella pneumoniae* were isolated from patients admitted to different departments at Assiut University Hospital over a period of 14 months, from April 2018 to May 2019. *Klebsiella pneumoniae* were isolated from endotracheal aspirate samples, sputum samples, blood samples, urine samples, wound swabs, and throat swabs. Isolates were identified by an API20E kit (BioMerieux, Marcy L Etoile. France).

### 4.2. Antimicrobial Susceptibility Testing

The antimicrobial susceptibility of the isolated *K. pneumoniae* was tested by the Kirby-Bauer disc diffusion method according to recommendations of the clinical laboratory standards institute [27]. The following 18 commercial antimicrobial discs were used: Aztreonam (30 μg), Amoxicillin (10 μg), Amoxicillin-Clavulanic acid (2010 μg), Piperacillin (100 μg), Ceftriaxone (30 μg), Ceftazidime (30 μg), Cefoperazone (75 μg), Cefpodoxime (10 μg), Meropenem (10 μg), Imipenem (10 μg), Trimethoprim/Sulfamethoxazole (1.25/23.75 μg), Gentamycin (10 μg), Amikacin (30 μg), Nitrofurantoin (300 μg), Levofloxacin (5 μg), Tetracycline (30 μg), Cefoxitin (30 μg), and Cefepime (30 μg). The *E. coli* ATCC 25922 strain was used for quality control. The test was repeated twice for each isolate. In addition, the minimum inhibitory concentrations (MIC) of imipenem against the tested *K. pneumoniae* were tested using Imipenem E-test strips (bioMérieux, Solna, Sweden).

### 4.3. Detection of Carbapenemase-Producing Isolates

#### 4.3.1. Modified Carbapenem Inactivation Methods (mCIM) for the Suspected Carbapenemase Producing Isolates 

Modified carbapenem inactivation methods (mCIM) were performed as described by Pierce et al. [44] for suspected Carbapenemase Producing *K. pneumoniae* isolates. The inhibition zones for meropenem were determined after incubation at 35 °C for 18 or 24 h. A test is considered negative for carbapenemase production if the zone diameter is ≥19 mm and is considered positive if the zone diameter is 6 to 15 mm or features the appearance of pinpoint colonies within a 16 to 18 mm zone. 

#### 4.3.2. EDTA-Modified Carbapenem Inactivation Method (eCIM) for Detection of MβL Enzymes 

If the mCIM test is positive, the EDTA-modified carbapenem inactivation method (eCIM) should be applied to differentiate between metallo-β-lactamases (MBL; Class B carbapenemases) and serine carbapenemases (Class A and D carbapenemases) in the *K. pneumoniae* isolates.

Bacterial isolates showing a positive mCIM test were cultured on Trypticase soy agar with sheep blood (TSAB). The EDTA-modified carbapenem inactivation method (eCIM) was performed according to Sfeir et al. [45]. The isolates were considered positive for MBL production if the zone diameter of meropenem disc increased by ≥5 mm in comparison to the zone diameter observed for the mCIM and were considered negative for MBL production if the increase in the zone diameter was ≤4 mm.

### 4.4. Detection of Hypermucoviscocity Phenotyping

The hypermucoviscocity phenotypes of the CPKP strains were checked by using the string test. The formation of a viscous filament of ≥5 mm was observed after stretching a Klebsiella spp colony with a loop cultured on an agar plate. CPKP strains positive on string tests may be considered hypervirulent CPKP (hvCPKP) [46].

### 4.5. PCR for Carbapenemase Genes and KI and K2 Genes 

*K. pneumoniae* isolates were tested by PCR for the detection of carbapenemase genes. The primers used in this study were tested for *bla-_NDM1_, bla-_KPC2_, bla-_VIM1_, bla-_IMP1_*, and capsular type (*wzy_K1* and *wzy_K2*) genes, as described previously by Poirel et al. [35]; Fang et al. [47]; Yigit et al. [18]; and Shibata et al. [48].

### 4.6. Plasmid Transfers

Conjugation experiments were performed with modifications according to Hardiman et al. [17]. For broth culture mating, the donor and recipient (*E. coli* J-53, azide resistant strain) cultures were mixed at a 1:4 ratio in fresh TSB and incubated at 37 °C without agitation for 20 h. The selection of transconjugants was done on a MacConkey agar containing azide (100 μg/mL) and meropenem (0.5 μg/mL). Transconjugants were tested against the same antibiotic discs used against the donor isolates. Then, the transconjugants were tested for *bla-_NDM1_* and/or *bla-_KPC2_* by PCR, as mentioned above. 

### 4.7. PCR-Based Replicon Typing (Inc/Rep PCR) for Major Plasmid Incompatibility Groups among Klebseilla Isolates and Transconjugant DNA Lysates

Fourteen pairs of primers were obtained to perform PCR-based replicon typing (PBRT), in which the multiplex PCR for FII, FIA, and FIB was performed as previously described by Carattoli et al. [49], while 11 simplex PCRs were performed for IncL, IncM, IncT, FIC, FIIK, IncN, IncX3, IncH12, IncW, IncY, and IncA/C as described by Carattoli et al. [50]; Carattoli et al. [49]; Villa et al. [51]; Johnson et al. [52]; García-Fernández et al. [16]; and Carloni et al. [53]. 

## 5. Conclusions

Molecular typing of plasmids based on replicon typing showed that Inc FIIK and FII plasmids were predominant among isolates in our area and their transconjugants carrying *bla_KPC-2_* and/or *bla_NDM-1_*. Also, conjugative Inc FII plasmids were found to have an important role in the spread of CPKP, and their recognition is essential to limit their spread.

## Figures and Tables

**Figure 1 antibiotics-08-00266-f001:**
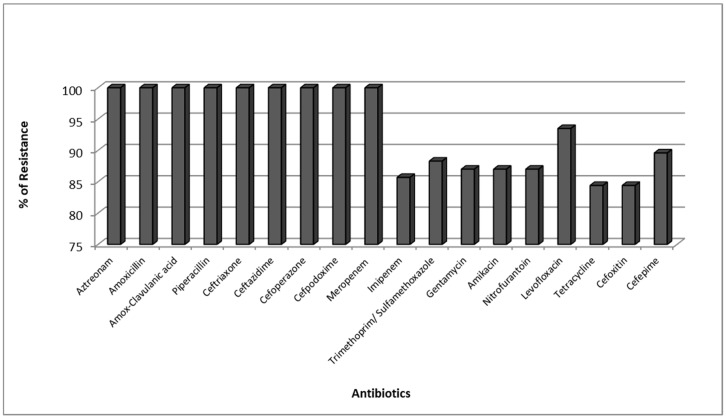
Resistance Pattern of Carbapenemase producing *K. pneumoniae* (CPKP).

**Figure 2 antibiotics-08-00266-f002:**
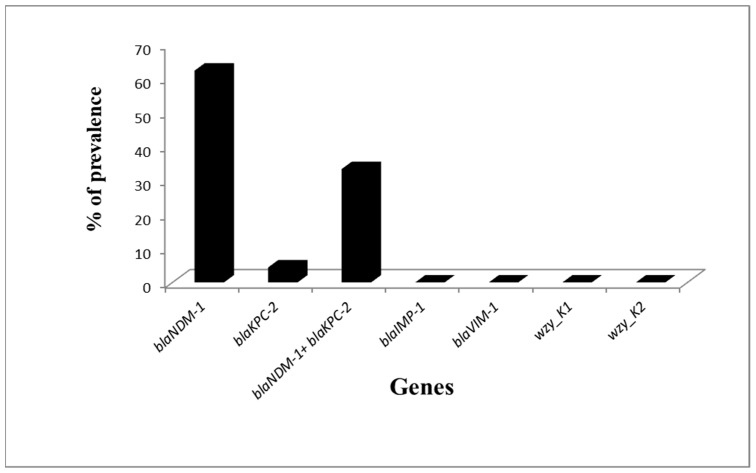
Prevalence of Carbapenemase genes and capsular genes among Carbapenemase producing *K. pneumoniae* (CPKP).

**Table 1 antibiotics-08-00266-t001:** Phenotypic and genotypic methods for the detection of CPKP.

Genotypic Method	No. of Isolates	Phenotypic Detection of Carbapenemase Producing Isolates	
		mCIM No. (%) *	eCIM No. (%) *
**Group 1**	32	0 (0)	NT
**Group 2**	28	28 (100)	9 (32.1)
**Group 3**	2	2 (100)	2 (100)
**Group 4**	15	15 (100)	5 (33.3)

* Percentages were correlated to the number of isolates of each group. Group 1: Negative for all tested genes; Group 2: positive for *bla-_NDM-1_* and negative for *bla-*_KPC-2_, *bla-*_IMP-1_ and *bla-*_VIM-1._ Group 3: positive for *bla-*_KPC-2_ and negative for *bla-*_IMP-1_, *bla-*_VIM1_ and *bla-_NDM-1_.* Group 4: positive for *bla-*_KPC-2_ and *bla-_NDM-1_* and negative for *bla-*_IMP-1_, *bla-*_VIM1._ NT: Not tested, mCIM: modified carbapenemase inhibition method, eCIM: EDTA-modified carbapenemase inhibition method.

**Table 2 antibiotics-08-00266-t002:** Resistance patterns of the 12 isolates and transconjugates showing different resistance profiles.

	Antibiotics	Tetracycline	Gentamicin	Sulfamethoxazole/Trimethoprim	Levofloxacin
Strains					
KP3/transconjugate		R/S	R/S	R/S	R/R
KP4/transconjugate		R/S	R/S	R/R	R/R
KP5/transconjugate		R/R	R/S	R/S	R/R
KP7/transconjugate		R/S	R/S	R/R	R/R
KP9/transconjugate		R/S	R/S	R/R	R/R
KP10/transconjugate		R/S	R/S	R/R	R/R
KP15/transconjugate		R/S	R/R	S/S	R/R
KP16/transconjugate		R/R	R/R	R/S	R/R
KP19/transconjugate		R/S	R/R	R/R	R/S
KP25/transconjugate		R/R	S/S	R/S	R/S
KP28/transconjugate		R/R	R/S	R/S	R/R
KP29/transconjugate		R/R	R/S	R/R	R/R

KP: *K. pneumoniae*, R: resistant, S: sensitive.

**Table 3 antibiotics-08-00266-t003:** Relation between the plasmid replicon type and the transfer of resistant genes.

Strain Code	Carbapenemase Gene	PBRT of Isolates	PBRT of Transconjugant
Kp1	NDM-1, KPC-2	FII	FII
KP2	NDM1	FIIK	FIIK
KP3	NDM-1	FII, FIIK, L	FIIK
KP4	NDM-1	M, L	-
KP5	NDM-1	-	-
KP6	NDM-1, KPC-2	-	-
KP7	NDM-1	FIB, M	-
KP8	KPC-2	FII, FIIK, FIB	FII, FIIK
KP9	NDM-1	FIIK, FIB	FIIK, FIB
KP10	NDM-1	FIIK, M	FIIK
KP11	NDM-1	FIC	-
KP12	NDM-1	FIIK, FIB, FII, M	FIIK, FIB
KP13	NDM-1, KPC-2	FIC	-
KP14	NDM-1	FIB, FIIK, FII, FIC	FIIK
KP15	NDM-1	FIC	-
KP16	NDM-1, KPC-2	FII, FIIK, FIB	FII, FIIK, FIB
KP17	KPC-2	FII, FIIK	-
KP18	NDM-1, KPC-2	FII, FIIK, FIB	FIIK
KP19	NDM-1	L	-
KP20	NDM-1	FIB	-
KP21	NDM-1, KPC-2	FII, FIIK, FIB	FII, FIIK, FIB
KP22	NDM-1	-	-
KP23	NDM-1, KPC-2	FII, FIIK, FIB, M	FII, FIIK, FIB
KP24	NDM-1, KPC-2	FIIK, FII, FIB,	FIIK
KP25	NDM-1, KPC-2	FII, FIIK	FII, FIIK
KP26	NDM-1	FIIK, FIB, L, M	FIB
KP27	NDM-1	L, M	-
KP28	NDM-1	FIC, FII, FIIK	FII
KP29	NDM-1, KPC-2	FIIK, FII	-
KP30	NDM-1, KPC-2	FIIK, FII, FIB	FIIK
KP31	NDM-1, KPC-2	FIIK, FII, M	FIIK
KP32	NDM-1	FIIK, FIC, FIB	FIIK

KP: *K. pneumoniae*, KPC: Carbapenemase-producing *K. pneumoniae*, NDM: New Delhi-metallo beta-lactamase.

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
