# Peer review of "The Inc FII Plasmid and its Contribution in the Transmission of *bla_NDM-1_* and *bla_KPC-2_* in *Klebsiella pneumoniae* in Egypt"

_antibiotics, 2019, doi:10.3390/antibiotics8040266_

Round 1

Reviewer 1 Report

Comments to the Author

Manuscript ID: antibiotics-658983

Title: Inc FII plasmid and its contribution in transmission of blaNDM-1 and blaKPC-2 in Klebsiella pneumoniae in Egypt.

The work presented has a great interest in getting to know more about the infections transmitted in hospital facilities. But the presentation of results and the objectives of the study can be improved

About the novelty of the content, the authors have not made clear the purpose of locating the plasmid that is transferred between any of the strains of K. pneumoniae, and how this may affect its control.

A propos of the potential impact of the manuscript in the relevant field in toxinology, in the article it has been shown that through the process of conjugation between bacteria they can survive extreme conditions. In my opinion, it would be necessary to justify how the transfer process can be redirected towards the control of the problem in hospital.

It would be convenient to improve the methodology section, in section 4.2 (line 207-215) the authors have not left reflected the antibiotic units that were used in the experiment. Also, the don´t say how many times have they repeated the antimicrobial susceptibility test.

At the introduction, the authors could talk about the origin and implications of K. in the health of the patients.

General and specific comments:

Line 63-64, what implications does the result have on Hypermucoviscocity phenotyping?

Line 140, change “cused” by “caused”

In the table 4- the authors could reflect what the abbreviations used in it mean. I did not find what NDMI was referring to

Reviewer 2 Report

The manuscript “Inc FII plasmid and its contribution in transmission of blaNDM-1 and blaKPC-2 in Klebsiella pneumoniae in Egypt” covers a topic of importance and is of overall interest. The authors appear to have done pioneering work by conceptualizing and demonstrating the emergence of blaKPC-2 and blaNDM-1 producing Klebsiella pneumoniae in many Egyptian hospitals.

The discussion is overall brief and shallow, it would benefit by more rigorous discussion. The author reference possible limitations of blaNDM-1 and blaKPC-2 in Klebsiella pneumoniae due to complications in multi-drug resistance study in hospitals, but should consider the potential pathophysiological aspects and implications.
